# GOLT1B Activation in Hepatitis C Virus-Infected Hepatocytes Links ER Trafficking and Viral Replication

**DOI:** 10.3390/pathogens11010046

**Published:** 2021-12-31

**Authors:** Jacqueline Butterworth, Damien Gregoire, Marion Peter, Armando Andres Roca Suarez, Guillaume Desandré, Yannick Simonin, Alessia Virzì, Amal Zine El Aabidine, Marine Guivarch, Jean-Christophe Andrau, Edouard Bertrand, Eric Assenat, Joachim Lupberger, Urszula Hibner

**Affiliations:** 1Institut de Génétique Moléculaire de Montpellier, University of Montpellier, CNRS, 34293 Montpellier, France; jackie@wordperfectscience.com (J.B.); marion.peter@inserm.fr (M.P.); guillaume.desandre@igmm.cnrs.fr (G.D.); yannick.simonin@umontpellier.fr (Y.S.); amal.makrini@igmm.cnrs.fr (A.Z.E.A.); jean-christophe.andrau@igmm.cnrs.fr (J.-C.A.); Edouard.bertrand@igh.cnrs.fr (E.B.); e-assenat@chu-montpellier.fr (E.A.); 2Inserm, U1110, Institut de Recherche sur les Maladies Virales et Hépatiques, 67000 Strasbourg, France; armando-andres.roca-suarez@inserm.fr (A.A.R.S.); virzi@unistra.fr (A.V.); marine.guivarch@etu.unistra.fr (M.G.); joachim.lupberger@unistra.fr (J.L.); 3Université de Strasbourg, 67000 Strasbourg, France; 4Department of Hepatogastroenterology, Hepatology and Liver Transplantation Unit, Saint Eloi Hospital, University of Montpellier, 34000 Montpellier, France

**Keywords:** hepatitis C, transcriptomic profiling, hepatocellular carcinoma, smiFISH

## Abstract

Chronic hepatitis C carries a high risk of development of hepatocellular carcinoma (HCC), triggered by both direct and indirect effects of the virus. We examined cell-autonomous alterations in gene expression profiles associated with hepatitis C viral presence. Highly sensitive single molecule fluorescent in situ hybridization applied to frozen tissue sections of a hepatitis C patient allowed the delineation of clusters of infected hepatocytes. Laser microdissection followed by RNAseq analysis of hepatitis C virus (HCV)-positive and -negative regions from the tumoral and non-tumoral tissues from the same patient revealed HCV-related deregulation of expression of genes in the tumor and in the non-tumoral tissue. However, there was little overlap between both gene sets. Our interest in alterations that increase the probability of tumorigenesis prompted the examination of genes whose expression was increased by the virus in the non-transformed cells and whose level remained high in the tumor. This strategy led to the identification of a novel HCV target gene: GOLT1B, which encodes a protein involved in ER-Golgi trafficking. We further show that GOLT1B expression is induced during the unfolded protein response, that its presence is essential for efficient viral replication, and that its expression is correlated with poor outcome in HCC.

## 1. Introduction

Chronic hepatitis C predisposes patients to life-threatening pathologies, including hepatocellular carcinoma (HCC) [1]. The persistence of the virus and the accompanying inflammation-driven liver disease are major contributors to the progression from steatosis to fibrosis, cirrhosis, and ultimately HCC. In addition, cell-autonomous effects of the virus that disturb cellular homeostasis and generate a pro-oncogenic environment via deregulation of hepatic metabolism and signal transduction further increase the risk of tumorigenesis (for review see [2,3,4]).

In the case of hepatitis C virus (HCV), the distinction between direct and indirect effects of the virus on the host cell is not clear-cut. Indeed, HCV-infected hepatocytes secrete a number of active molecules that impact the liver physiopathology through both autocrine and paracrine signaling. In addition to the well-studied cases of class I interferon [5], TGFβ and VEGF [6,7], lymphotoxin β, and wnt-mediated signaling that originate from the HCV-harboring cells also belong to this group [8,9]. An unbiased genome-wide proteogenomic approach highlighted major HCC hallmarks that are induced by HCV infection, including EGFR, STAT3, epithelial–mesenchymal transition (EMT), and perturbations in liver metabolism and DNA repair [10,11]. Interestingly, key HCV-dysregulated signaling pathways are also major players in the tight regulation of liver regeneration [12]. Importantly, even when viral infection is cleared, an epigenetic imprinting prevents a full recovery and contributes to the elevated HCC risk observed in HCV cured patients [13,14,15]. Despite recent progress in our understanding of the virus–host interactions, much remains to be learned about subtle virus-driven cell-autonomous alterations that may escape detection in a bulk analysis of the “omics” landscape. 

In order to identify HCV-positive and HCV-negative hepatocytes, we used an improved version of single molecule fluorescent in situ hybridization methodology (smiFISH) [16] for frozen tissue sections from a chronic hepatitis C patient. Mapping of virus-infected cells was followed by tissue microdissection and transcriptomic profiling, leading to the identification of candidate gene products deregulated by HCV. One such deregulated gene discovered in our study is GOLT1B, encoding a so far poorly-studied protein involved in endoplasmic reticulum (ER)-Golgi trafficking [17,18]. We further show that GOLT1B is involved in the unfolded protein response (UPR) and is essential for HCV replication.

## 2. Results

### 2.1. HCV RNA Detection by Single Molecule In Situ Hybridization

Single molecule fluorescent in situ hybridization (smFISH) has previously been used to detect and quantify HCV viral RNA within infected hepatocytes in frozen liver sections of patients with hepatitis C [19]. However, the initial procedure is both time-consuming and expensive; indeed, the technique requires the use of long (~50 bp) specific probes covering about 1 kb of the target RNA sequence [20]. Given the high sequence divergence in different HCV subtypes [21], each patient’s viral isolate needs to be sequenced prior to probe design. Patient-specific sets of fluorescent probes are then synthesized and used for FISH experiments. In contrast, in the recently developed highly sensitive and versatile version of single molecule FISH, the smiFISH [16], the target-specific sequence of each probe is shorter (26–32 nucleotides) and a single secondary detection probe is used, thus considerably lowering the cost of the experiment and allowing the use of a large number of probes for the targeted RNA (reviewed in [22]). We reasoned that it should be possible to design a set of probes with a broader specificity, recognizing, if not all HCV isolates, then at least all viral RNAs within a given genotype. In this work, we used a consensus sequence derived from 20,000 genotype 1b sequences deposited in the euHCVdb database (http://euhcvdb.lyon.inserm.fr/euHCVdb/; accessed on 26 March 2016) to design 61 probes, the locations of which are shown in Figure 1A (and Appendix A). We predict that every viral genome should be recognized by a minimum of 40 probes. 

The sensitivity and specificity of the pan-genotype 1b HCV probes were first tested on Huh7 Nneo/c-5B cells that harbor the full-length genotype 1b replicon [23]. As shown in Figure 1B, a strong signal was detected in the replicon cells, but not in the parental Huh7 line. The Nneo/c-5B replicon cells contain a high concentration of HCV RNA, visualized by hundreds of dots upon smiFISH analysis (Figure 1B). The control probe set, specific for the Firefly luciferase mRNA, gave no signal in either cell line (Figure 1C).

In order to further test the sensitivity of our detection procedure, we next explored detection of the signal in frozen liver sections from a HCV transgenic FL-N/35 mouse model, expressing very low levels of viral RNA [24] and thus reminiscent of the clinically relevant situation found in livers of chronically infected hepatitis C patients. Cytoplasmic HCV RNA was clearly detectable in transgenic mouse livers and absent from the control wild type livers (Figure 1D).

### 2.2. Detection of HCV-Infected Cells in Human Liver Sections

High variability of the viral load and of the fraction of infected hepatocytes in livers of hepatitis C patients has been reported [19,25]. Tumors arising in the context of hepatitis C have been described to have low viral levels [26,27,28], in coherence with the diminished expression of miR-122, an essential co-factor of viral replication (reviewed in [29]). To further investigate this point, we designed pan-genotype HCV primers to compare the abundance of viral RNA in tumoral and peritumoral samples of a small cohort of 20 patients (Figure 2A). As expected, our data show important inter-patient variation and confirm a lower viral load in the majority of tumors, as compared to corresponding non-tumoral tissue. Interestingly however, in at least four out of 20 tumors, the viral load was higher than in the surrounding tissue. We have chosen to concentrate further analysis on patient #8, who displayed high viral abundance (genotype 1b) both within the tumor and in the non-tumoral tissue. 

Pan-HCV genotype 1b smiFISH probes were used on OCT-frozen liver tissue from patient #8, a 62-year-old male patient who had undergone HCC resection on cirrhotic liver with signs of chronic hepatitis C. Confirming the RT-qPCR analysis (Figure 2A), a clear HCV-specific signal was observed in both the tumoral and peri-tumoral tissue (Figure 2B). Many tumor cells gave a very strong HCV-specific signal, visualized as cytoplasmic aggregated dots (arrowheads), considerably larger than the size expected from a single RNA molecule detection (arrows). These may represent sites of intense RNA replication, further arguing for a strong productive infection of the tumoral tissue. As previously reported, the HCV-infected cells were mainly visible as clusters (see e.g., [19]), allowing laser microdissection and capture of HCV-positive and HCV-negative regions from the frozen liver specimen. 

### 2.3. Gene Expression Profiling of HCV-Positive and HCV-Negative Cell Clusters 

We used smiFISH-labelled tissue sections from patient #8 to delineate HCV-positive and negative regions. For this, we examined confocal tile scan images combining the smiFISH data for the tissue sections representing average areas of 2.5 cm^2^, corresponding to circa 10,000 cells. Clusters of infected and HCV-free hepatocytes were identified by visual examination (Figure 3A). A total of 11 HCV-positive and 7 HCV-negative regions, each composed of 100–300 cells, were used as the dissection guide on serial tissue sections surrounding these tile scan images for laser microdissection, RNA extraction, and RNAseq analyses. Indeed, since the smiFISH procedure is not compatible with good quality RNA preparation, the microdissection was performed on the adjacent cuts of serial sections used for the smiFISH imaging. We used 10-µm-thick sections. Because hepatocytes are large cells of about 40 µm diameter, we reasoned that adjacent serial sections will largely have the same pattern of HCV+ and HCV− cell clusters. Transcriptomic profiling confirmed the correct identification of HCV presence or absence in 70% of these samples. All of the discordant results were false positives, i.e., they concerned samples initially identified as HCV+; these were omitted from further analysis. The normalized HCV read values in the confirmed positive samples varied between 0.3–4.2 for the peritumoral and between 1.1 and 24.2 for the tumoral clusters (Figure 3B).

A total of eight samples from the tumor and four from the peri-tumoral tissues were thus subjected to further analysis by RNAseq. We identified 202 genes that were differentially expressed in peritumoral tissue as a function of viral presence (137 up and 65 down, *p* < 0.05, log2 fold change > 2) (Figure 3C). In the tumor, the total number of dysregulated genes was 257 (98 up, 159 down) (Figure 3D) (listed in Appendix A). For the upregulated genes, GO analysis indicated strongest enrichment of the “Golgi membrane” category for the non-tumoral samples (fold enrichment = 3.1, *p* = 1.5 × 10^−3^) and “metal ion binding” and “DNA binding” for the tumoral part (fold enrichment = 2.2 for both, *p*-value = 1.1 × 10^−3^ and 6.2 × 10^−3^, respectively) (Appendix A).

Somewhat unexpectedly, we found very little overlap between genes that appeared dysregulated by HCV in the peritumoral and in the tumoral tissues. This negative result made us question the initial assumption that HCV-driven transcriptional changes would be apparent independently of the cellular context, in this case either the tumoral or the peritumoral tissue. 

### 2.4. Expression of a Subset of HCV Dysregulated Genes Is Increased during Tumorigenesis 

We reasoned that some alterations in gene expression caused by the virus are likely to play a role in tumor development [30,31] and their expression might be maintained in the tumor even if the initial stimulus, i.e., the virus, is no longer present. Of note, there is a precedent for such phenomena, both in the “hit-and-run” strategy of some oncogenic viruses and in the case of stable epigenetic changes following HCV eradication [14,32,33,34]. As a consequence, we sought to identify genes that are dysregulated in the HCV+ non-tumoral samples and whose expression remains dysregulated in the tumor, independently of the HCV status of the analyzed region. Twenty-five genes that were upregulated in the HCV+ areas and ten whose expression was decreased in the presence of the virus fell into this new category (Figure 3E). Our data reveal the presence of several cancer-related pathways in this small gene set, namely, ECM interactions and cytoskeleton dynamics (ARHGAP5, CADM1, CHI3L1, ITGA6, TMOD3, ATXN1L, MYO19), metabolism and oxidative stress (SUCLA2, ME1, OSER1, SLC35B1, TTC19, SLC5A9, HCCS), proliferation signaling (CCND2, STK38, RASSF4), circadian clock (METTL14, CLOCK, KDM8), and vesicle trafficking (SNX2, GOLT1B, AP4E1). We were particularly interested in this latter category since it is related to the most highly enriched gene set (Golgi membrane) detected by the GO analysis of peri-tumoral samples. In order to pinpoint the genes most likely to correspond to bona fide downstream transcriptional targets of HCV, we next concentrated on the upregulated genes and sought to correlate the level of expression of these putative targets with the level of HCV RNA present in the tissue. To do so, we performed RT-qPCR analysis on 10 candidates on peri-tumoral and corresponding tumoral regions in the 10 patients of the original cohort for whom we detected viral RNA (Figure 2A). For two out of 10 candidates, we could indeed detect a strong correlation between the viral presence and the gene’s expression in the non-tumoral tissues (Figure 4A and Appendix A). In contrast, in the tumors, the expression remained high, but was no longer correlated with the viral load. One such gene was CLOCK, a central component of the circadian clock, which regulates several steps in the HCV life cycle, including particle entry into hepatocytes and RNA genome replication [35]. Another gene in this category was GOLT1B, an evolutionary conserved gene encoding a protein involved in vesicular Golgi trafficking [17,36]. Interestingly, analysis of an independent cohort of HCV-infected patients without liver tumors [37] revealed a correlation between CLOCK and GOLT1B expression as well as a correlation between both genes with ER stress and UPR gene sets (Figure 4B and Appendix A). 

### 2.5. GOLT1B Is Required for Efficient HCV Replication 

Because deficiencies in GOLT1B homologues in yeast and in rice (Got1p and Glup2 genes, respectively) disturb redistribution of proteins from the ER to the Golgi [17,18], we asked if GOLT1B might be associated with ER stress and a subsequent UPR in hepatocytes. This proved to be the case, since pharmacological induction of ER stress and UPR by thapsigargin in Huh7.5.1 cells gave rise to a significant increase of GOLT1B expression (Figure 5A). 

We next asked if increased GOLT1B expression in HCV-infected cells was a mere reflection of an overall ER stress or whether this protein had a functional importance in the viral life cycle. Efficient GOLT1B silencing by siRNA in Huh7.5.1 cells gave rise to only a minor effect on cell viability (Figure 5B). Control and GOLT1B KD cells were then infected with reporter viruses encoding Firefly luciferase. We tested viral entry by using HCV pseudoparticles (HCVpp) that display the HCV envelope glycoproteins on the backbone of a retroviral vector, as well as the entry and the replication of the bona fide HCV viral construct (HCVcc). While GOLT1B deficiency had no effect on viral entry, replication efficiency was significantly compromised in GOLT1B KD cells (Figure 5C). Thus, GOLT1B plays a role in the HCV life cycle. 

We next questioned GOLT1B involvement in hepatic tumorigenesis. Kaplan–Meier analysis of TCGA data indicated that a high level of GOLT1B expression was associated with poor overall survival for virus-related HCC (Figure 5D) but had no significant impact on the survival of patients with many other tumor types, such as colorectal carcinoma, invasive breast cancer or glioblastoma. Moreover, GOLT1B hepatic expression was significantly higher in patients with severe fibrosis score compared to low fibrosis. Finally, in support of our conclusions regarding the role of GOLT1B in hepatic tumorigenesis, its expression was significantly higher in the tumor compared with non-tumoral tissues of both hepatitis B and hepatitis C patients (Figure 5E).

## 3. Discussion

Despite remarkable progress in prevention and treatment, chronic infections with HBV or HCV remain the major risk factors for HCC [41]. In addition to necro-inflammatory liver damage, characteristic of chronic viral hepatitis and responsible for creating a favorable environment for tumor development, direct effects of the virus on the host cell have been incriminated as HBV-induced pro-oncogenic events. These events include insertional mutagenesis, transcriptional deregulation or inactivation of the p53 tumor suppressor by a viral protein [42]. The case is less clear for HCV, although several transgenic mouse models expressing all or a subset of the HCV proteins are tumor-prone, strongly arguing for their oncogenic activity in the absence of any immune-mediated hepatic lesions [24,43]. Moreover, transcriptional and post-transcriptional activation of oncogenic signaling pathways, as well as inhibition of apoptosis, has been described for several viral proteins [9,10,44,45]. Importantly, both viruses trigger long-lasting alterations of the epigenome of their host cells [15], which may account for a proportion of HCC cases developing after the efficient clearance of HCV infection [44,45].

The lack of animal models recapitulating the events leading from chronic hepatitis C to tumorigenesis is a major hurdle for a full understanding of the underlying mechanisms. However, recent technological developments render a feasible in-depth analysis of surgical samples from liver resection of patients suffering from HCC that developed on the HCV infected liver. Here, we report such an effort that has led to the characterization of virus-mediated changes in the transcriptomic profile occurring in the true physiopathological context of naturally occurring hepatitis C infection.

There are two options for comparing the transcriptional profiles of patient-derived cells that either harbor the virus or are free of it. Single cell RNA sequencing is one of them. While extremely powerful, this technique has two major disadvantages: its high cost and the fact that it provides no positional information on the cells in the sample. In contrast, RNAseq analysis of microdissected regions, classified as infected or virus-free by single molecule FISH imaging, allows the study of infected cell clusters and their comparison with cells that are not in their immediate vicinity. This last point may be of importance because of the documented effects of infected cells on their non-infected counterparts, likely to primarily operate over short distance. Moreover, in situ hybridization and microdissection can be performed on frozen tissue samples allowing a retrospective study of appropriately preserved samples.

Driven by these considerations, we employed an improved version of smFISH: the single molecule inexpensive (smi)FISH [16]. Importantly, while smiFISH retains the high sensitivity of the classical smFISH, it is more versatile and considerably cheaper. This is due to the use of short unlabeled primary oligonucleotide probes that are tagged by a common sequence. Thus, a single fluorescent-labelled oligonucleotide, which is complementary to the common tag, is needed for in situ hybridization. An additional improvement came from designing a set of oligonucleotide probes that are expected to recognize all of the known sequences of the genotype 1b HCV. Of note, although we have not employed smiFISH for detection of any other low-abundance RNA viruses, there are no theoretical obstacles for doing so.

The RNAseq analysis of samples microdissected from clinical specimens remains technically challenging, and the quantity and quality of the recovered RNA did not allow high-depth analysis. It is therefore very likely that our dataset represents only a subset of alterations in gene expression triggered by the viral presence. Indeed, the number of genes dysregulated in the course of in vitro viral infection is 5 to10-fold higher than that detected in our work [10]. Noteworthy, however, our methodology is expected to strongly enrich the identification of cell-autonomously regulated genes and it has been performed on clinical samples from a chronic hepatitis C patient. Further analyses of additional patient samples will be required to better define the sets of genes commonly deregulated by HCV in a clinically relevant setting.

There are several common selective pressures that must be dealt with in a persistent viral infection and in a growing tumor, such as novel metabolic requirements and escape from elimination. This is presumably why some mechanisms used by viruses to pervert cellular functions constitute a risk factor for transformation [30,31]. Following this line of thought, we have discovered genes whose expression is dysregulated in infected, non-transformed cells as well as in tumor cells, independently of their infection status. These genes seem specifically related to an environment of advanced liver disease and HCC since they do not overlap with previous transcriptomics from HCV infection models [10]. This analysis led to the identification of GOLT1B, an ER protein involved in ER-Golgi protein trafficking [17,18]. GOLT1B has not previously been reported in the context of HCV infection, and we confirm that it is not directly induced by HCV, in accordance with previous omics studies on Huh7.5.1 cells and human liver chimeric mice [10]. We show, however, that the basal GOLT1B expression greatly facilitates HCV replication, presumably in relation to the major role played by ER in the viral life cycle. Indeed, HCV infection gives rise to major expansion and reorganization of the ER, leading to the creation of a membranous web, which is the site of viral replication and assembly (reviewed in [46]). This is achieved through the action of non-structural proteins NS4B and NS5A and is accompanied by the activation of the cellular response to stress, the unfolded protein response (UPR). The UPR is a common cellular adaptation to numerous stresses, originally described as a survival mechanism allowing cells to deal with an overload of protein processing in the ER (reviewed in [47]). Because viral or bacterial infection as well as cancer leads to an increased demand on protein synthesis and processing, these conditions are often associated with the UPR [48,49]. Interestingly, the UPR has also been described for other pathological conditions, including hepatic pathologies, such as NAFLD, fibrosis, and cirrhosis [50,51], which constitute independent risk factors for HCC. 

Importantly, the UPR is not only a consequence of a strong protein synthetic activity in rapidly growing cells, it is also a necessary adaptation for cancer cell survival and growth [52]. Similar to many other cellular stress responses, overwhelming UPR can also trigger cell death, and novel anti-tumor therapies currently under investigation aim either at inhibiting or augmenting UPR processes. 

While the general mechanism of the UPR is shared by many cell types and many types of stress, the response is fine-tuned by the preferential use of its three main sensors and many partners engaged in complex signaling networks [53]. It remains to be investigated if GOLT1B is involved in the generic UPR or rather remains specific to hepatic physiopathology. Similarly, further work is necessary to establish whether the requirement for GOLT1B during HCV infection reflects a need for UPR-mediated ER reorganization or if this new HCV target gene plays a novel distinct role in the viral life cycle.

## 4. Material and Methods

### 4.1. Cells

Huh7 cells were cultured in DMEM supplemented with 10% fetal bovine serum, 100 µg/mL streptomycin, and 100 U/mL penicillin; 400 µg/mL of G418 was added to Huh7 Nneo/C-5B cells that harbored the full-length HCV genotype 1b replicon [23]. 

### 4.2. Mice

Three-month-old FL-N/35 males [24], transgenic for the entire HCV genotype 1b open reading frame and wild type controls, both in the C57Bl/6J genetic background, were used. Mice were housed and bred according to French Institutional guidelines. The protocols were approved (ID of approval for this study N° F 34-172-16) by the Languedoc-Roussillon ethics committee (CEEA-LR1013). 

### 4.3. Patients

Freshly frozen tumoral and peritumoral tissue from 20 cases of HCC surgical resection on hepatitis C background were collected from Montpellier University Hospital. Two additional patients had HCC unrelated to HCV. The samples were anonymized and used after obtaining written informed consent from patients, in accordance with French legislation.

### 4.4. HCV Infection

Huh 7.5.1 cells [54] were infected with HCV-derived pseudoparticles (H77; genotype 1a) or cell culture-derived infectious HCV (HCVcc; strain Luc-jc1) harboring a luciferase reporter gene, as previously described [55]. Virus entry and infection were assessed two days after infection by measuring reporter gene luciferase activity in cell lysates using the Bright Glo Luciferase assay system (Promega, Charbonnières-les-Bains, France) and a Mithras LB 940 luminometer (Berthold Technologies, Bad Wildbad, Germany).

### 4.5. HCV Pan-Genotype 1b smiFISH Probe Design

More than 20,000 HCV genotype 1b sequences >9.5 Kb in length from the NCBI database were aligned using Galaxy [56]. A consensus HCV genotype 1b sequence was generated in Aliview (the Open Source Software License ‘GNU General Public License, version 3.0 (GPLv3)). In the event of missing sequence information, the non-identified nucleotides were arbitrarily designated as C. Probes were designed by R-script Oligostan [16], which automatically eliminates C-stacks and thus excludes any artificially created C-blocks. Sixty-one primary probes corresponding to the HCV genotype 1b consensus sequence were designed. The same software was used to design primary probe sets for negative controls, the Firefly luciferase and Hygromycin resistance genes, as well as the positive controls, i.e., mouse and human GAPDH. 

### 4.6. smiFISH Applied to Fixed Cells

The reagents used, primary probe preparations, FLAP probe sequence, and preparations, as well as the smiFISH protocol were carried out according to the detailed description found in [16]. Briefly, a set of target-specific probes was synthesized, each carrying an additional 28-nt long sequence (“FLAP”), which is not represented in either mouse or human genomes. Cy-3 labelled fluorescent probe complementary to FLAP was also synthesized and pre-annealed to primary probes prior to in situ hybridization. 

### 4.7. smiFISH Applied to Frozen Tissue 

Freshly dissected human or mouse liver tissue fragments were frozen in OCT in liquid nitrogen-cooled isopentane and stored at −80 °C. Ten-µM-thick tissue sections were mounted on Superfrost^TM^ Plus Gold slides (Thermo Fisher Scientific, Waltham, MA, USA), fixed in 4% paraformaldehyde pH 9 for 30 min at room temperature, and permeabilized for 30 min in 1% Triton/PBS at room temperature. SmiFISH was performed as above using RNAse-free glass coverslips. Nuclei were counterstained by ProlongGold DAPI (Thermo Fisher Scientific, Waltham, MA, USA).

### 4.8. RNA Extraction and RT-qPCR

RNA was extracted from frozen samples of either tumoral or matched non-tumoral parts of HCC patient biopsies using the RNeasy mini kit (Qiagen, Hilden, Germany); 0.25 μg of total RNA was reverse-transcribed with the QuantiTect Reverse Transcription kit (Qiagen), and cDNA was quantified using LC Fast start DNA Master SYBR Green I Mix (Roche, Basel, Switzerland). Primers for RT-qPCR were designed using LightCycler probe design 2 software (Roche), targeting the pan-genotype small (i.e., 300 bp) conserved region in the 5′UTR of HCV viral sequence: forward: 5′- CAGGAGATGGGCGGAAAC -3′, reverse: 5′- GCCGCAATGGATATTTCATTCTCA-3′. Results were normalized to SRSF4 housekeeping gene expression; forward: 5′-CGGAGTCCTAGCAGGCATA-3′, reverse: 5′-TTCCTGCCCTTCCTCTTGT-3′. GOLT1B expression was quantified using qPCR primers GOLT1B-fw (5′-CGGCTTCATTTCTCCCGACT-3′) and GOLT1B-rv (5′-TCCAATTTTCTGCGTGTCCG-3′) using the SYBR green method using a CFX96 Touch Thermal Cycler (Bio-Rad, Hercules, CA, USA). PCR products were validated by melting curve analysis and Sanger sequencing.

### 4.9. Microscopy and Imaging

In vitro smiFISH microscopy was performed using a widefield Zeiss Axioimager Z1 to capture Z-stacks using an X63 1.4 NA oil objective equipped with a CCD camera (Axiocam) and controlled with Metamorph (Molecular Devices). Tissue smiFISH imaging was performed using an inverse SP8-UV confocal microscope (Leica) for Z-stack capturing with a X63 1.4 NA oil objective and HyD detectors. Z-stacks and confocal tile scans were compiled using LAS software (Leica Application Suite). Signals were detected after adjustment of the negative control probe to zero (considered background noise). The remaining non-specific signal, likely due to spectral bleed-through, was calculated by image overlying upon excitement of the Cy3 and GFP channels. For the HCV signal the non-specific component was estimated to be 1.8% of the total signal and was therefore considered negligible. Estimation of the fraction of HCV-positive cells in tissue sections was performed by manual counting of HCV-positive cells relative to the total number of detected DAPI-stained nuclei in five randomly chosen fields (>100 cells/field). 

### 4.10. Laser Microdissection and RNA Extraction

Serial 10 µM sections of human liver tissue were mounted on PET-membrane steel frame slides (Leica) and stored at −80 °C until use. For laser dissection, slides were fixed for 20 min in 70% ethanol at +4 °C followed by dissection using a PALM MicroBeam laser UV dissector (Zeiss, Oberkochen, Germany). Serial section alignment of the smiFISH and unstained slides was carried out at the Laser Microdissection platform at Bordeaux Neurocampus using an automated serial section alignment and microdissection (PALM 4 technology, Zeiss). RNA was extracted and purified from each sample (*circa* 100–200 cells) using the RNeasy^®^ Micro kit (Qiagen, Hilden, Germany) with on-column DNAse I digestion. RIN values were calculated using the Agilent Bioanalyzer with RNA pico chips. 

### 4.11. RNAseq 

RNA samples containing a minimum of 1 ng RNA and a RIN > 4 were used for further RNAseq analysis. First, RNA was amplified using the Ovation^®^ SoLo RNA-seq library preparation kit (NuGEN) used for low input or single-cell sequencing according to manufacturer’s instructions, including a ribosomal RNA depletion step (Insert Dependent Adaptor Cleavage). Libraries were then monitored for concentration and fragment sizes using a Fragment Analyzer (kit Standard Sensitivity NGS) and by qPCR (ROCHE Light Cycler 480) prior sequencing on Illumina HiSeq2000 platform (single end 50 bp length).

### 4.12. Single Sample Gene Set Enrichment Analysis

Transcriptomic data from HCV-infected non-treated human liver samples (GSE84346, *n* = 22) were preprocessed with the CollapseDataset tool available at GenePattern. Single sample gene set enrichment analysis (ssGSEA) was performed using gene sets belonging to the Molecular Signatures Database (MSigDB) version 7.4 [38,39]. The correlations of GOLT1B with CLOCK mRNA levels and ER/UPR with circadian rhythm signaling pathway scores were assessed with Pearson’s correlation coefficient.

### 4.13. Statistical Analysis

Read quality assessment

The first five bases of each read were trimmed using cutadapt v. 1.13 (option -u 5). The trimmed reads were aligned to the human genome (Hg38) using TopHat2 v2.1.1 [57]. The read quality was controlled using FastQC and FastQ Screen tools.

Identification of viral reads

In order to confirm the presence of the hepatitis C virus in HCV-positive samples, we aligned the RNA reads on the HCV sequences using the BWA mapper tool (using BWA-backtrack algorithm). The results confirmed the presence of HCV in seven samples. Based on this result, we removed five HCV-positive samples for which no reads mapped on the HCV reference genome. We also removed one sample considered HCV-negative but that mapped reads to the HCV reference. 

Differential Gene Expression analysis

Differential Gene Expression analysis was performed using the DESeq package [58] from Bioconductor. First, the featureCounts program (version 1.6.2) was used to count the reads that mapped to gene annotations with the option ‘-s 1’. Then these counts were analyzed using the DESeq package to identify genes that were at least 1-fold (log2) differentially expressed relative to the reference sample using a *p*-value threshold of 0.05.

Statistical analysis of GOLT1B experiments was performed using GraphPad prism software (GraphPad Software, San Diego, CA, USA).

## Figures and Tables

**Figure 1 pathogens-11-00046-f001:**
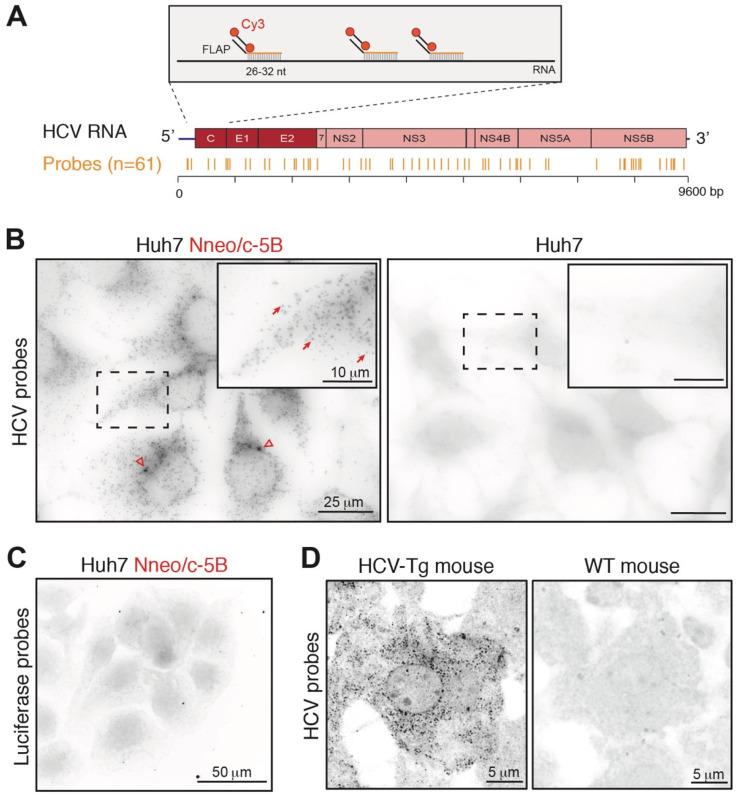
Detection of HCV RNA by smiFISH. (**A**) Principle of smiFISH. Localization of the 61 probes along viral RNA is indicated. (**B**) HCV RNA detection by smiFISH performed on Huh7-Nneo/c-5B replicon and Huh7 control cells. Arrows indicate single RNA molecules, arrowheads point to aggregates likely to correspond to replication sites (**C**) SmiFISH with luciferase control probe set on Huh7 Nneo/c-5B replicon cells (**D**) SmiFISH detection of HCV RNA on liver sections from HCV FLN-35 transgenic and control WT mice.

**Figure 2 pathogens-11-00046-f002:**
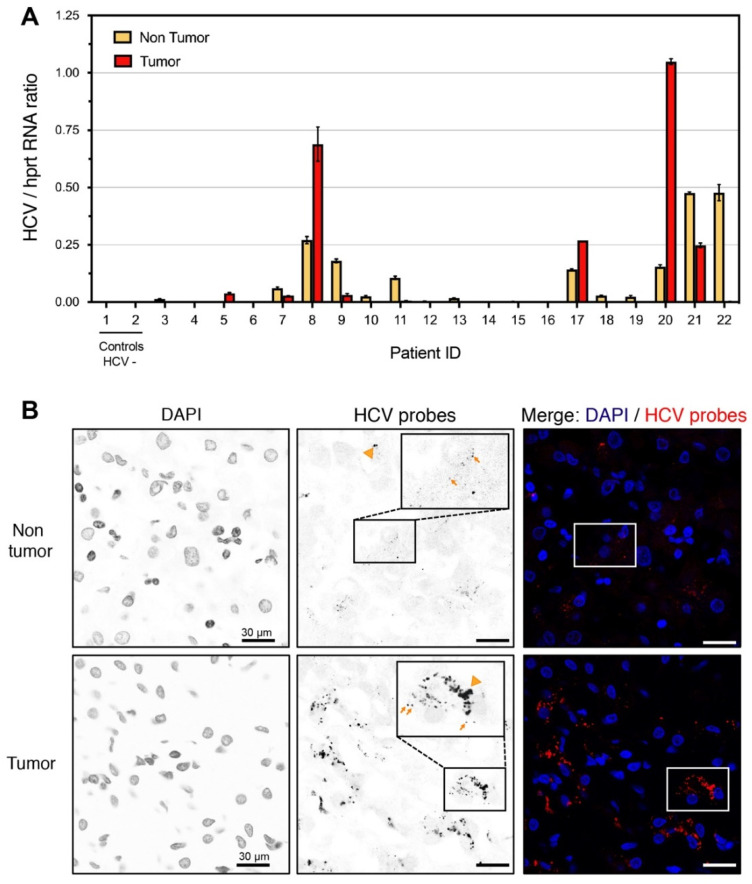
Detection of HCV RNA by smiFISH in a patient with HCC. (**A**) Quantification of HCV RNA by RT-qPCR in non-tumoral and tumoral tissues of HCC patients. Patients #1 and #2 were negative for HCV and were used as controls. Means +/-SD of technical triplicates are shown. **(B**) smiFISH on patient#8 non tumoral and tumoral regions. Arrows indicate examples of single viral RNA molecules; arrowheads point to aggregates that may correspond to active viral replication sites.

**Figure 3 pathogens-11-00046-f003:**
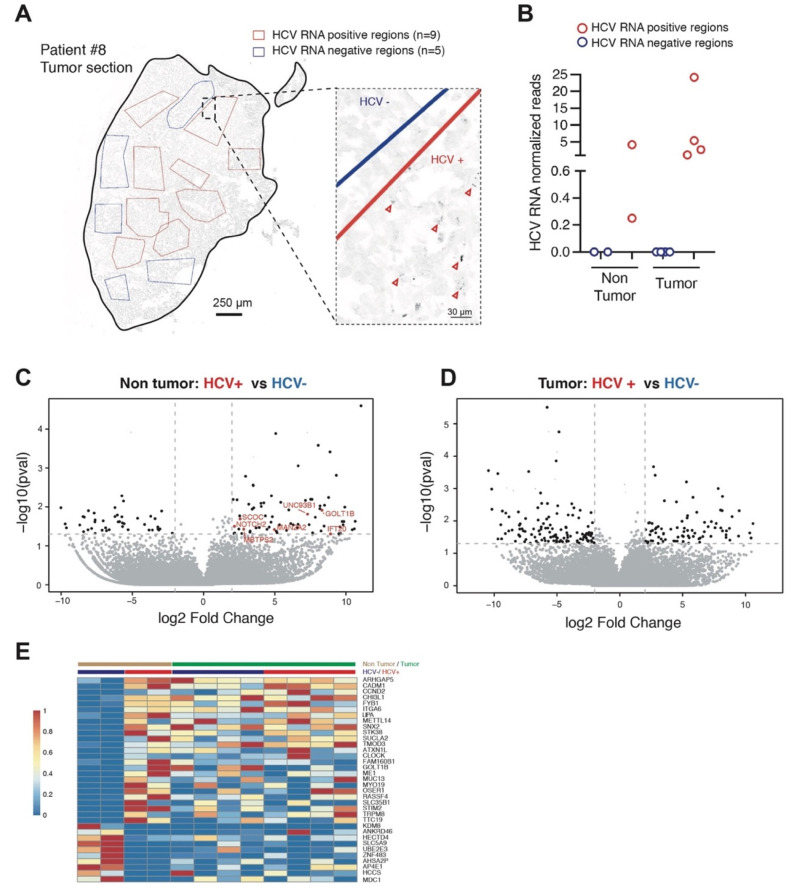
Transcriptomic analysis of HCV+ and HCV- clusters in tumoral and non-tumoral parts of a HCC patient. (**A**) An example of a tile scan image used to identify HCV-positive (red) and -negative (blue) regions detected by smiFISH that was used as guide for laser microdissection on serial tissue sections. Arrowheads on zoomed insert show examples of smiFISH-HCV positive cells. (**B**) HCV reads in RNAseq transcriptomes of HCV+ and HCV- regions used for further analysis. HCV reads were normalized to the total number of reads and number of cells in dissected regions. (**C**) Volcano plot presentation of genes deregulated in non-tumoral HCV+ vs. HCV- cells. (**D**) Volcano plot presentation of genes deregulated in tumoral HCV+ vs. HCV- cells. (**E**) Heatmap of genes deregulated in HCV+ non-tumoral samples that remained deregulated in the tumor.

**Figure 4 pathogens-11-00046-f004:**
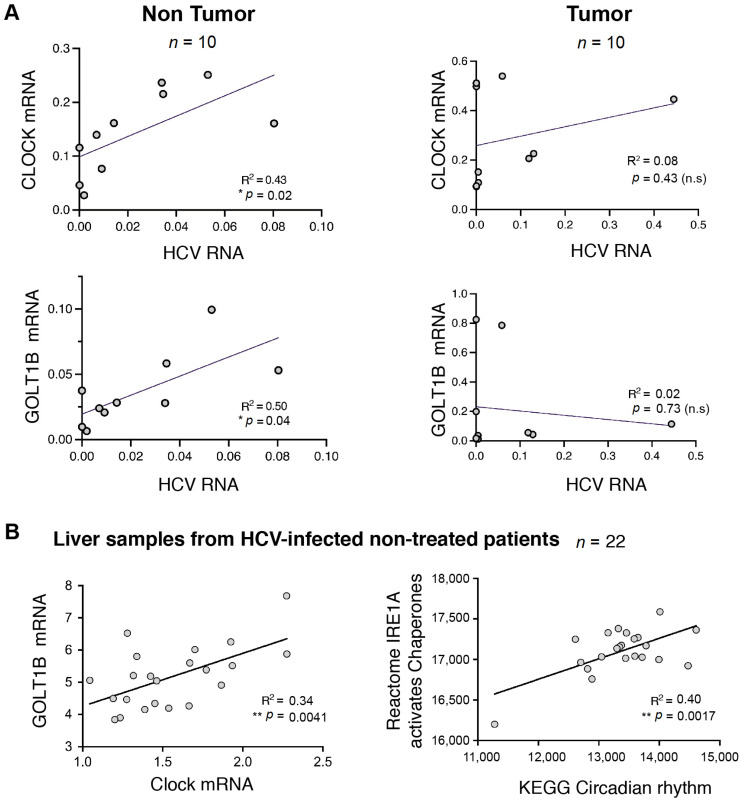
Correlation of CLOCK and GOLT1B expression with HCV RNA levels in non-tumoral liver in HCC patients. (**A**) mRNA levels of CLOCK/GOLT1B were quantified by qPCR in non-tumoral and tumoral samples from ten patients with HCC of HCV etiology. The R-squared and *p*-values from Pearson correlation tests (two-tailed) are indicated. (**B**) ER/UPR signaling pathways are associated with the circadian rhythm in the liver. The association of GOLT1B with CLOCK mRNA levels and the association of ER/UPR with the circadian rhythm signaling pathway were assessed in liver samples from HCV-infected non-treated patients (GSE84346, *n* = 22; [37]). Reactome and KEGG gene sets were obtained from the Molecular Signature Database (MSigDB) [38,39] are blotted as enrichment scores (ES); mRNA expression is plotted as normalized reads per kilo base per million mapped reads (RPKM). R-squared and *p*-values from Pearson correlation tests (two-tailed) are indicated. * *p* < 0.05, ** *p* < 0.01.

**Figure 5 pathogens-11-00046-f005:**
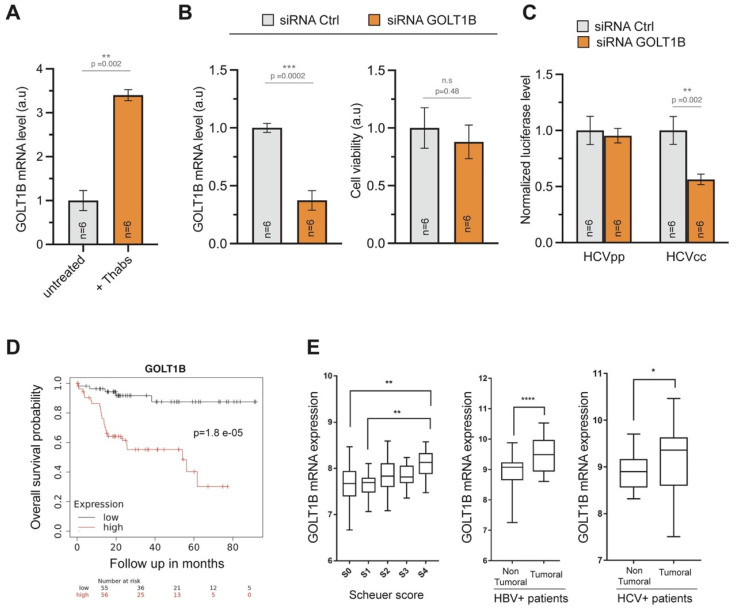
GOLT1B is an essential factor in HCV replication. (**A**) The Unfolded Protein Response (UPR) significantly increases GOLT1B expression in Huh7.5 cells treated for 8 h with 1 µM thapsigargin (Thabs). Results are displayed as average GOLT1B mRNA expression relative to GAPDH +/− SD (three biological replicates in technical duplicates). The *p*-value from the Mann–Whitney U-test test is indicated. (**B**) GOLT1B silencing efficacy in Huh7.5.1. (left panel). Measurement of cell viability in an HCVpp-infected cell using PrestoBlue (right panel). Means +/− SEM are shown. The Mann–Whitney U-test statistical significance is indicated. (**C**) Assessment of viral entry and replication via HCV pseudoparticles (HCVpp) and HCV viral construct (HCVcc). siGOLT1B impairs viral replication but not viral entry (mean +/− SEM, Mann–Whitney, U-test). (**D**) Kaplan–Meier analysis of TCGA data indicates that a high GOLT1B expression is correlated with poor overall survival probability in patients with HCC associated with viral hepatitis. Analysis conducted using Kaplan–Meier Plotter [40] (**E**) GOLT1B mRNA expression is significantly higher in HBV patients with severe fibrosis than in patients with mild fibrosis, according to the histological staging of fibrosis (Scheuer score “S”). In total, 124 liver biopsy samples were retrieved (GEO accession number: GSE84044) and used for the bioanalysis (S0 = 43 patients, S1 = 20 patients, S2 = 33 patients, S3 = 18 patients, S4 = 10 patients). For statistical analysis, the Kruskal–Wallis test (non-parametric ANOVA) was performed and GP *p*-values calculated: 0.0021 (**), GOLT1B expression in whole liver tissue was analyzed from 39 samples from HBV-associated HCC patients “Non tumor area (HBV)” and 81 samples from HBV-associated HCC patients “Tumor area (HBV)”. The samples derived from 11 HBV-associated HCC patients who underwent liver transplantation for tumor (GEO accession number: GSE107170). GOLT1B expression in whole liver tissue was analyzed from 31 samples from HCV-associated HCC patients “Non tumor region (HCV)” and 44 samples from HCV-associated HCC patients “Tumor region (HCV)”. The samples derived from 11 HCV-associated HCC patients who underwent liver transplantation for tumor (GEO accession number: GSE107170). For statistical analysis, the Mann–Whitney test was performed and GP *p*-values were calculated: >0.05 (not significant, n.s.), <0.05 (*), <0.01 (**), <0.001 (***), <0.0001 (****).

## Data Availability

RNA-sequencing data has been deposited in the Gene Expression Omnibus (GEO, NCBI) repository and are accessible through GEO Series accession number GSE192862.

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
