# Peer review of "GOLT1B Activation in Hepatitis C Virus-Infected Hepatocytes Links ER Trafficking and Viral Replication"

_pathogens, 2021, doi:10.3390/pathogens11010046_

Round 1

Reviewer 1 Report

Although a very effective standard of care is available for Hepatitis C treatment there are still cases where the liver pathology can not be reversed after viral clearance. The molecular mechanisms which lead to HCC in chronically infected patients are still not completely understood. Herein, Butterworth et al. applied the smiFISH technique to identify HCV RNA in hepatocytes from frozen liver sections. Transcriptomic analysis on tissue regions harvested by laser microdissection was used to identify endogenous factors potentially involved in the viral induced transformation of the infected hepatocytes. The authors identified GOLT1B as a gene upregulated in non-tumoral tissue depending on HCV RNA presence which remained overexpressed in tumoral tissue but independent of HCV RNA. GOLT1B gene expression was further shown to correlate with HCV RNA levels in non-tumoral liver tissue and poor survival for HCC. GOLT1B expression was correlated with the liver fibrosis stage in HBV positive patients. By siRNA downregulation, the authors show that GOLT1B is required for HCV replication, but not entry.

This work is quite welcomed in the field because it presents a tool to identify viral induced tumorigenesis factors using hepatocytes in their biological environment rather than whole liver tissue or cell culture systems.

Minor corrections

Although it is a proof of concept on one patient, it would be interesting to comment on the overlap with the hit lists obtained in the past from acute infections in HCVcc system or infected uPA/SCID or human liver tissue samples (e.g. Diamond et al., Plos Pathog.,2011; Lupberger et al., Gastroenterology, 2019).  

At line 248 it is stated that “GOLT1B hepatic expression is significantly higher in patients with low fibrosis score compared to severe fibrosis”. On the other hand, in Figure 5E it is shown a direct correlation between GOLT1B expression and liver fibrosis stage for HBV positive patients. The authors should correct in the text and address the correlation between GOLT1B expression and the fibrosis stage for HCV positive patients also.  For Kaplan-Meier analysis in Figure 5D, it would be informative to know the number of patients HCV positive and HBV positive, respectively.

Author Response

We thank the reviewers for their comments, which we believe helped us to improve our manuscript. The point-by-point response can be found below and the changes made in the text are highlighted in red in the revised version of the manuscript. We believe we have addressed all the reviewers' queries and hope that our manuscript will be judged of value for publication in Pathogens.

Reviewer 1

Although a very effective standard of care is available for Hepatitis C treatment there are still cases where the liver pathology cannot be reversed after viral clearance. The molecular mechanisms which lead to HCC in chronically infected patients are still not completely understood. Herein, Butterworth et al. applied the smiFISH technique to identify HCV RNA in hepatocytes from frozen liver sections. Transcriptomic analysis on tissue regions harvested by laser microdissection was used to identify endogenous factors potentially involved in the viral induced transformation of the infected hepatocytes. The authors identified GOLT1B as a gene upregulated in non-tumoral tissue depending on HCV RNA presence which remained overexpressed in tumoral tissue but independent of HCV RNA. GOLT1B gene expression was further shown to correlate with HCV RNA levels in non-tumoral liver tissue and poor survival for HCC. GOLT1B expression was correlated with the liver fibrosis stage in HBV positive patients. By siRNA downregulation, the authors show that GOLT1B is required for HCV replication, but not entry.

This work is quite welcomed in the field because it presents a tool to identify viral induced tumorigenesis factors using hepatocytes in their biological environment rather than whole liver tissue or cell culture systems.

We thank the reviewer for his/her appreciation of our study.

Minor corrections

Although it is a proof of concept on one patient, it would be interesting to comment on the overlap with the hit lists obtained in the past from acute infections in HCVcc system or infected uPA/SCID or human liver tissue samples (e.g. Diamond et al., Plos Pathog.,2011; Lupberger et al., Gastroenterology, 2019).

We agree with the reviewer that this is an interesting point that we did not discuss in the original version of the manuscript. In fact, we performed this analysis and did not find any overlap of the genes identified in the current study with the previously reported datasets from RNA-seq and proteomic analyses of HCV-infected Huh7.5.1 and uPA/SCID mice (Lupberger et al., Gastroenterology 2019). Importantly, in contrast to the patient-derived samples, the available experimental HCV infection models and datasets do not induce liver fibrosis or HCC. This major difference in the physiopathological context of the infection suggests that the genes identified in the present study are induced in a setting of advanced liver disease and HCC, but probably not directly by the HCV infection per se. We now added a sentence mentioning this finding in the discussion of the revised manuscript on line 325.

At line 248 it is stated that “GOLT1B hepatic expression is significantly higher in patients with low fibrosis score compared to severe fibrosis”. On the other hand, in Figure 5E it is shown a direct correlation between GOLT1B expression and liver fibrosis stage for HBV positive patients. The authors should correct in the text and address the correlation between GOLT1B expression and the fibrosis stage for HCV positive patients also.

We apologize for the error in the sentence. Indeed, GOLT1B expression is significantly lower in samples from patients with low fibrosis score compared to severe fibrosis, as correctly shown for hepatitis B patients in Fig. 5E. The mistake has now been corrected in the manuscript. The scarcity of transcriptomic datasets from HCV-infected patient livers with corresponding liver fibrosis staging does not allow a statistical valid conclusion. Indeed, the only dataset available from Markus Heims group (Boldanova et al., EMBO Mol Med. 2017, PMID 28360091) comprises very few patients with F3/F4. Thus, we are not able to provide a statement about a potential association of GOLT1B expression during HCV-associated liver fibrosis.

For Kaplan-Meier analysis in Figure 5D, it would be informative to know the number of patients HCV positive and HBV positive, respectively.

In the revised manuscript, we have now refined the survival analysis and excluded patients with alcohol abuse. It now focuses only on HCCs associated with viral hepatitis. The new figure 5D also indicates also the number of patients at risk.

Reviewer 2 Report

In this manuscript, by employing semi-quantitative in-situ hybridization, microdissection and RNA-seq techniques, the authors investigated gene expression altered by HCV infection in hepatocytes. The findings are interesting, but somewhat preliminary. The following concerns need to be addressed.
  1. Please show the complete lists of pathways enriched by the GO analyses.
  2. The expression of GOLT1B and ER stress should be determined in HuH7.5.1 infected with HCV.
  3. ER stress should be examined in HuH7.5.1 cells knocked down for GOLT1B.
  4. line 192: The authors say that 10 genes were analyzed. The results of the other 8 genes should be shown.
  5. CLOCK gene and circadian rhythm are suggested to associate with ER stress. Is there a possible association between CLOCL and GOLT1B during HCV infection?

Author Response

Reviewer 2

In this manuscript, by employing semi-quantitative in-situ hybridization, microdissection and RNA-seq techniques, the authors investigated gene expression altered by HCV infection in hepatocytes. The findings are interesting, but somewhat preliminary. The following concerns need to be addressed.

We thank the reviewer for his/her appreciation of our study.

  1. Please show the complete lists of pathways enriched by the GO analyses.

The list of pathways enriched is now provided in new Supplementary Table 3.

  1. The expression of GOLT1B and ER stress should be determined in HuH7.5.1 infected with HCV.

As suggested by the reviewer, we now provide a new Supplementary Figure S2 demonstrating an HCV-induced enrichment of a gene set comprising genes related to ER stress and UPR (HALLMARK_UNFOLDED_PROTEIN_RESPONSE) in HCV-infected Huh7.5.1 and human liver chimeric mice at the RNA and protein levels (Lupberger et al., Gastroenterology 2019, PMID 30978357). These data are in accordance with previous reports in the literature (Sir et al., Hepatology 2008, PMID 18688877; Tardif et al., Trends Microbiol. 2005, PMID 15817385).

Following the reviewer's request, we have investigated GOLT1B expression and ER stress- and UPR-related gene set enrichments in HCV-infected Huh7.5.1 and in human liver chimeric mice (Lupberger et al., Gastroenterology 2019, PMID 30978357). However, no significant change of GOLT1B expression was observed in either of the two experimental models (results mentioned in the revised version of the manuscript line 330). This is in perfect agreement with our interpretation stating that GOLT1B is induced in a setting of advanced liver disease and HCC, but not directly by HCV infection per se (see the response to the first point raised by the reviewer #1 and a sentence added to the revised version of the manuscript: discussion, line 325).

  1. ER stress should be examined in HuH7.5.1 cells knocked down for GOLT1B.

ER stress and UPR are known to be induced by HCV infection. However, as mentioned above, GOLT1B expression appears to be induced by HCV only in the context of advanced liver pathology. We have therefore not expected alterations in ER stress and or the UPR in Huh7.5.1 cells silenced for GOLT1B. We have, however, performed the experiment requested by the reviewer using the extracts from the experiment shown in Fig. 5B. We chose the markers CHOP (DDIT3), ATF3, and ATF6, which are significantly induced by HCV infection in Huh7.5.1 (Lupberger et al., Gastroenterology 2019). As predicted, we did not observe an induction of ER stress/UPR after GOLT1B knock down (figure 1 to reviewer#2), thus strengthening our interpretation that GOLT1B acts downstream of UPR.

Please see the attached word document for access to the figure.

Figure 1 to reviewer 2: GOLT1B silencing in Huh7.5.1 does not impact ER stress/UPR marker gene expression. RNA extracts from Fig. 5B have been analyzed for CHOP, ATF3, and AFT6 expression and are normalized to GAPDH expression. Relative expression is expressed as the mean relative expression of three biological replicates in triplicates. Ns=not significant according to an unpaired T-test (Parametric-passed normality test). 

  1. line 192: The authors say that 10 genes were analyzed. The results of the other 8 genes should be shown.

The results for the 8 other genes are now shown in Supplementary Figure 1.

  1. CLOCK gene and circadian rhythm are suggested to associate with ER stress. Is there a possible association between CLOCK and GOLT1B during HCV infection?

Indeed, we observe a positive correlation between CLOCK and GOLT1B and gene sets of circadian rhythm in livers of HCV-infected, in patients who have not received anti-viral treatment (Boldanova et al., EMBO Mol Med. 2017). We provide this analysis in the revised Fig. 4B of the manuscript and we added a sentence on line 207.

Round 2

Reviewer 2 Report

The authors have sufficiently addressed my concerns.